# Impact of Reaction Parameters and Water Matrices on the Removal of Organic Pollutants by TiO_2_/LED and ZnO/LED Heterogeneous Photocatalysis Using 365 and 398 nm Radiation

**DOI:** 10.3390/nano12010005

**Published:** 2021-12-21

**Authors:** Máté Náfrádi, Tünde Alapi, Gábor Bencsik, Csaba Janáky

**Affiliations:** 1Department of Inorganic and Analytical Chemistry, University of Szeged, Dóm tér 7, H-6720 Szeged, Hungary; nafradim@chem.u-szeged.hu; 2Department of Physical Chemistry and Materials Science, University of Szeged, Rerrich Béla tér 1, H-6720 Szeged, Hungary; bencsikg@chem.u-szeged.hu (G.B.); janaky@chem.u-szeged.hu (C.J.)

**Keywords:** hydroxyl radical, carbonate radical, matrix effect, coumarin, neonicotinoid

## Abstract

In this work, the application of high-power LED_365nm_ and commercial, low-price LED_398nm_ for heterogeneous photocatalysis with TiO_2_ and ZnO photocatalysts are studied and compared, focusing on the effect of light intensity, photon energy, quantum yield, electrical energy consumption, and effect of matrices and inorganic components on radical formation. Coumarin (COU) and its hydroxylated product (7-HC) were used to investigate operating parameters on the ^•^OH formation rate. In addition to COU, two neonicotinoids, imidacloprid and thiacloprid, were also used to study the effect of various LEDs, matrices, and inorganic ions. The transformation of COU was slower for LED_398nm_ than for LED_365nm_, but r_0_^7-HC^/r_0_^COU^ ratio was significantly higher for LED_398nm_. The COU mineralization rate was the same for both photocatalysts using LED_365nm_, but a significant difference was observed using LED_398nm_. The impact of matrices and their main inorganic components Cl^−^ and HCO_3_^−^ were significantly different for ZnO and TiO_2_. The negative effect of HCO_3_^−^ was evident, however, in the case of high-power LED_365nm_ and TiO_2_, and the formation of CO_3_^•−^ almost doubled the r_0_^7-HC^ and contributes to the conversion of neonicotinoids by altering the product distribution and mineralization rate.

## 1. Introduction

Advanced Oxidation Processes (AOPs) may offer a solution to remove trace amounts of organic pollutants from aqueous and gaseous media [1]. In the case of AOPs, in addition to efficiency, energy demand is also important, and the light source in photochemical processes determines both. Heterogeneous photocatalysis based on the irradiation of the appropriate semiconductor photocatalysts is a widely investigated method. The absorption of photons having higher energy than the bandgap of the photocatalyst leads to the formation of excited conduction band electrons (e_CB_^−^) and valence band holes (h_VB_^+^) [2]. In addition to recombination [3], the photogenerated charge carriers initiate the transformation of organic compounds via direct charge transfer on the surface of the catalysts or lead to the formation of various reactive oxygen species (ROS) [4]. The most important ROS is the hydroxyl radical (^•^OH) thanks to its high reactivity and low selectivity, resulting in fast transformation and adequate mineralization of organic pollutants during heterogeneous photocatalysis [2,5,6].

The two most widespread commercially available photocatalysts are TiO_2_ and ZnO, having relatively wide bandgaps; therefore, UV light is needed for their effective excitation. The bandgap reported for TiO_2_ is 3.0 eV for rutile and 3.2 eV for anatase phase; this value also varies between 3.1−3.3 for ZnO [7,8]. TiO_2_ has gained popularity due to its high activity, stability, chemical and biological inactivity, and relatively low price. ZnO is also widely investigated, as it has similar properties to TiO_2_, and lower production cost. The higher electronic conductivity of ZnO results in a faster charge transfer with the species on the surface and, consequently, lower recombination rates than TiO_2_ [9,10]. However, the photocatalytic properties of ZnO depend on the morphology and particle size [11,12,13], and its susceptibility to photo-corrosion limits its application [14,15].

Excitation of TiO_2_ and ZnO catalysts has traditionally been performed using different UV lamps (black lights, mercury vapor lamps, xenon lamps) [16]. Recently, Light Emitting Diodes (LEDs) have gained popularity thanks to new LEDs emitting in the UV-A region with high intensity, good electric efficiency [17], and several advantages compared to traditional UV sources, such as narrow wavelength ranges, which allow the construction of specialized photoreactors [17,18,19]. The UV-LED can be considered a quasi-monochromatic light source since the output energy is narrowly (λ_max_ ± 10 nm) distributed around the maximum of the wavelength. Depending on the wavelength used, they can have better electrical efficiency than traditional UV sources [20,21]. LEDs are also available at an affordable price while being compact and robust—thus, flexible and efficient photochemical setups can be constructed [16], and they have a significantly longer lifetime (>10,000 h). The application of UV-LEDs offers an alternative solution for the excitation of photocatalysts and has been employed for heterogeneous photocatalysis in recent years [20,22,23]. It is challenging to find the best design for photoreactors, such as optimizing the distribution and number of LEDs and light intensity [24].

For heterogeneous photocatalysis, the most important factors affecting the transformation rate of a given organic substrate and photocatalyst are the concentration of the substrate, the photocatalysts’ load, and the light intensity. In addition to these factors, the interaction between the substrate and photocatalysts’ surface, the properties of the photocatalysts, and the reactivity of the target substance towards possibly formed various reactive species are also significant [25]. The formation rate of photogenerated charges primarily depends on the light intensity and the absorption properties of the photocatalyst at the given wavelength [26,27,28]. The transformation rate of the target substances depends on the number of photogenerated charges migrated to the surface, which is highly limited by their recombination in bulk, which reduces the availability of photogenerated charges for redox reactions on the surface. It is found that the intensity (the photon flux or photon density) has a substantial impact on the lifetime of charge carriers [28,29,30]. The authors found that at low photon flux there is a linear relationship between the transformation rate and photon flux, while at high photon flux, the transformation rate exhibited a square root dependency on the light intensity [27,31]. However, the role and significance of wavelength [31,32,33,34] and “extra energy” of photons (the difference between the photon energy and bandgap energy) [10] for charge separation and recombination rate efficiency is rarely studied, and the results reported are relatively diffuse.

The efficiency of heterogeneous photocatalysis also depends on the properties of the treated water (e.g., pH, ionic content, dissolved organic matter). The matrix components may affect the surface properties of the photocatalyst, such as the surface charge and the adsorption of target pollutants, or act may as radical scavengers reacting with the photogenerated charges and ROS [35,36,37,38,39]. The role of inorganic ions is often discussed as radical scavengers [35,36,37]; their reaction with photogenerated charges is examined less frequently [38,40]. The fate of the radicals and/or radical ions originated from these inorganic ionic components of matrices, and their role and contribution to the transformation of target organic components during heterogeneous photocatalysis are not yet fully clarified. Our knowledge on the effect of matrices during AOPs is still limited; despite the high number of papers published on the topic, only a small fraction worked with actual wastewaters and even less investigated the effect of each matrix component in detail [41]. Another important shortcoming is that the publications focus on the effect on the conversion rate of the starting compound. However, the effect of the individual matrix components can balance each other, and the effects are manifested in the change of product distribution or the mineralization rate [41,42].

Coumarin (COU) and two neonicotinoids are used as target substances in this work. COU is used to compare ^•^OH formation rate in the case of heterogeneous photocatalysis [5,43,44,45,46,47,48,49,50,51]. Neonicotinoids are a class of insecticides causing severe environmental problems, the most well-known being their harmful effect on pollinators [52,53]. They can also have other adverse effects, such as toxicity on non-target organisms [54] or endocrine disruptive effects [55]. The use of several neonicotinoids, such as acetamiprid, imidacloprid (IMIDA), and thiacloprid (THIA) has been restricted by the European Union [56], but they are still extensively used worldwide. The efficiency of their photolytic removal varies; therefore, several AOPs have already been employed to treat neonicotinoid-containing waters, including homogeneous and heterogeneous photocatalysis [57,58,59,60].

This research aimed to compare the efficiency of commercial TiO_2_ and ZnO photocatalysts irradiated with a high-power UV-A LED (HP LED_365nm_) having a maximum emission of 365 nm and cheap, commercial LEDs with lower electric power consumption (LED_398nm_) having a maximum emission of 398 nm. The formation rate of the ^•^OH, the essential reactive species, was compared based on the formation rate of the hydroxylated product (7-hydroxy-coumarin (7-HC)) of coumarin (COU). The effect of photon flux, catalyst dosage, and COU concentration was investigated, and reaction parameters were optimized based on the rate and quantum yield of 7-HC formation. Two neonicotinoid pesticides, IMIDA and THIA, as environmentally relevant target substances, were chosen to compare the application of the TiO_2_ and ZnO photocatalysts and LEDs emitting 365 ± 10 nm and 398 ± 10 nm light. The transformation and mineralization rate, photonic efficiency, and electric power consumption required for the transformation were determined. The formation of organic products, inorganic ions, and change in the ecotoxicity during treatment was also measured. AOPs, including heterogeneous photocatalysis, can be used as an effective tertiary treatment for removing micropollutants, which remain in the water after the conventional physical-biological processes. The oxidation rate is affected by the dissolved organic matter and inorganic species of the water matrix. Accordingly, two water matrices were used in this work, biologically treated domestic wastewater with relatively high ionic and low organic content and tap water having lower ionic and organic content. Special attention was paid to studying the effect of Cl^−^ and HCO_3_^−^, and their combined effect as the main inorganic components of both matrices.

## 2. Materials and Methods

### 2.1. Photochemical Reactors and Light Sources

One of the photoreactors was equipped with high-power UV-A LEDs (Vishay; Malvern, USA; VLMU3510-365-130; LED_365nm_) emitting light from 355–380 nm, with UV-emission maximum at 365 nm. The 12 SMD diodes were soldered on metal core printed circuit boards (Meodex, Narbonne, France) and fixed on aluminum heat sinks (Fischer Elektronik; Lüdenscheid, Germany 0.70 K W^−1^). A laboratory power supply (Axiomet, Malmö, Sweden; AX-3005DBL-3; maximum output 5.0 A/30.0 V) was used to provide and control the electrical power needed to operate the light sources (P_el^max^_ = 21 W). The 200 cm^3^ solutions were irradiated in a cylindrical borosilicate [61] glass reactor, and the suspension was bubbled with gas (N_2_ (99.995%) or synthetic air) (Appendix A). 

The other photoreactor was equipped with UV-Vis LEDs (LED_398nm_) emitting light from 385–420 nm, with an emission maximum at 398 nm. The cheap, commercial LED_398nm_ tape (LEDmaster, Szeged, Hungary; P_el^max^_ = 4.6 W; 60 LED m^−1^) was fixed on the inner side of a water-cooled aluminum tube. The 100 cm^3^ solution was irradiated in a cylindrical borosilicate glass reactor (Appendix A). The electrical power of the LEDs was regulated by the power supply (100%, 50%, or 25% of P_radiant^max^_).

The emission spectra of the used light sources were measured using a two-channel fiber-optic CCD spectrometer (AvaSpec-FT2048, Avantes, The Netherlands) operated in the 180–880 nm wavelength range. The electric power consumption of LED_398nm_ was determined with a digital multimeter (Maxwell 25331; Oakland, CA, USA).

### 2.2. Photochemical Experiments and Analytical Methods

The photon flux of the light sources was measured by potassium-ferrioxalate actinometry, a standard one recommended by IUPAC [62] that is widely investigated and applied in the literature [62,63,64,65,66,67,68,69]. The 1.0 × 10^−2^ M Fe^3+^-oxalate solutions were irradiated, the released Fe^2+^ was measured after complexation with 1,10-phenanthroline. The absorbance of the Fe^2+^-phenanthroline complex was measured at 510 nm using UV-Vis spectrophotometry (Agilent 8453, Santa Clara, CA, USA) in a quartz cuvette with a 0.20 cm optical path length. The quantum yields applied for the calculation of the photon flux were slightly different: 1.21 for LED_365nm_ and 1.14 for LED_398nm_ [63].

During photocatalytic experiments, two commercially available photocatalysts were used, TiO_2_ Aeroxide P25^®^ (Acros Organics) and ZnO (d < 100 nm, Sigma Aldrich; St. Louis, MO, USA). Diffuse reflectance spectroscopy (DRS) was performed using an Ocean Optics USB4000 detector and Ocean Optics DH-2000 light source. The bandgap energy values of the photocatalysts were evaluated by the Kubelka–Munk approach and the Tauc plot.

The initial transformation rate of COU (r_0_^COU^) and the initial formation rate of 7-HC (r_0_^7-HC^) were determined from the linear part of the kinetic curves, up to 15% transformation of COU. Before analysis, the samples were centrifuged (Dragonlab; Beijing, China 15000 RPM) and filtered using syringe filters (FilterBio Nantong, China; PVDF-L; 0.22 µm). The COU concentration was measured using UV-Vis spectrophotometry at 277 nm (ε_277nm_ = 10293 M^−1^ cm^−1^). The concentration of COU in the treated sample was also determined by spectrophotometry and HPLC measurements. The difference between the determined concentrations for the same samples was less than 10%, so spectrophotometric determination was used for further experiments. The UV-Vis absorption and emission spectra of the model compounds are summarized in Appendix A. The concentration of the formed 7-HC was determined using fluorescence spectroscopy (Hitachi F-4500; Tokyo, Japan); the excitation and emission wavelengths were 345 nm and 455 nm, respectively. 

The concentration of IMIDA and THIA were determined by HPLC-DAD (Agilent; Santa Clara, CA, USA 1100, column: Lichrosphere 100, RP-18; 5 μm). The eluent consisted of 40 *v*/*v*% methanol (MeOH) and 60 *v*/*v*% water, the flow rate was 1.0 cm^3^ min^−1^, the temperature was set to 30 °C. Detection of IMIDA and THIA was performed at 270 nm and 242 nm, respectively; their retention time was 5.1 min and 9.1 min. The determination of the products was carried out by HPLC-MS measurements, with an Agilent LC/MSD VL mass spectrometer (Agilent, Santa Clara, CA, USA) coupled to the HPLC. The measurements were performed using an APCI ion source and a triple quadruple analyzer in positive mode (4000 V capillary voltage, 60 V fragmentor voltage, and 4.0 μA corona current). The flow rate of the drying gas was 4.0 dm^3^ min^−1^, and its temperature was 200 °C. The scanned mass range was between 50–500 AMU.

Total Organic Carbon (TOC) concentration was determined using an Analytik Jena (Jena, Germany) N/C 3100 analyzer. The formation of inorganic ions (Cl^−^, NO_2_^−^, NO_3_^−^, SO_4_^2−^ and NH_4_^+^) was measured using ion chromatography (Shimadzu, Kyoto, Japan) Prominence LC-20AD, Shodex 5U-YS-50 column for cation detection, and Shodex NI-424 5U for anion detection). The eluent for cations and anions was 4.0 mM methanesulfonic acid and a mixture of 2.5 mM phthalic acid and 2.3 mM aminomethane, respectively. The flow rate of the mobile phase was 1.0 cm^3^ min^−1^.

Ecotoxicity tests (LCK480, Hach Lange GmbH, Düsseldorf, Germany) were based on the bioluminescence inhibition of the marine bacteria *Vibrio fischeri*. H_2_O_2_, which form during the transformation of organic substances, was decomposed in the samples by adding catalase enzyme before starting the ecotoxicity tests. The catalase concentration in the samples was 0.20 mg dm^−3^. The bioluminescence of the test organism was measured using a Lumistox 300 (Hach Lange) luminometer after 30 min incubation time.

### 2.3. Chemicals and Solvents Used

Two commercial photocatalysts, TiO_2_ Aeroxid^®^ P25 (Sigma Aldrich; St. Louis, MO, USA) and ZnO (Sigma Aldrich; St. Louis, MO, USA, <100 nm) were used. The used TiO_2_ photocatalysts consist of 76–78% anatase phase and 10–16% rutile phase. The presence of amorphous TiO_2_ was also reported [70,71,72]. ZnO contains only wurtzite phase. The list of used chemicals can be found in Appendix A. The water matrices used were tap water (Szeged, Hungary) and biologically treated domestic wastewater (Szeged, Hungary); their parameters are summarized in Appendix A.

## 3. Results and Discussion

### 3.1. Photon Flux and Electrical Efficiency of the LEDs

The photon flux of both LEDs changes linearly with the electric power consumption (Appendix A) in the investigated range. For the photoreactor equipped with 12 pieces of LED_365nm_, the photon flux changed from 2.83 × 10^−6^ to 1.71 × 10^−5^ mol_photon_ s^−1^ when electric power increased from 3.4 W to 20.8 W. For LED_398nm_, the change of electric power from 0.96 W to 4.68 W increased the photon flux from 1.2 × 10^−6^ to 4.6 × 10^−6^ mol_photon_ s^−1^. In this case, 60 pieces of LEDs irradiated the reactor volume. The obtained electrical efficiencies (P_radiant_/P_electric_) were 27% for LED_365nm_ regardless of the photon flux, and 30–37% for LED_398nm_, decreasing with the increase of the photon flux (Appendix A).

### 3.2. Effect of Reaction Parameters on the ^•^OH Formation

The reaction between COU and ^•^OH (k_COU+•OH_ = 6.9 × 10^9^ M^−1^ s^−1^ [73]) radicals results in a highly fluorescent molecule, 7-hydroxycoumarin (7-HC). This method is fast and adequate with sufficient sensitivity, and it has been applied for detecting ^•^OH radicals produced in a photocatalytic system by many authors [5,43,44,45,46,47,48,49,50,51,73,74,75]. Thus, the optimization of reaction parameters was based on the formation rate and quantum yield of 7-HC formation for both light sources (LED_365nm_ and LED_398nm_) and photocatalysts (TiO_2_ and ZnO). The effect of the fundamental parameters, such as photocatalyst and COU concentration, and the photon flux was investigated. The relative adsorbed amount of COU and 7-HC was less than <1.0% in each case.

The photon flux was adjusted to a similar value (5.52 × 10^−6^ mol_photon_ s^−1^ for LED_365nm_ and 4.68 × 10^−6^ mol_photon_ s^−1^ for LED_398nm_) when the effect of the initial COU concentration and catalyst dosage were studied. The effect of photocatalyst dosage (0–1.5 g dm^−3^) was determined at 5.0 × 10^−4^ M COU concentration. For LED_365nm,_ over 0.5 g dm^−3^, the transformation rate of COU (r_0_^COU^) slightly increased or did not change significantly (Figure 1a). The r_0_^COU^ values determined for ZnO exceeded that determined for TiO_2_; however, the 7-HC formation (r_0_^7-HC^) was faster for TiO_2_ than for ZnO, indicating a more efficient ^•^OH formation. A plausible explanation of this fact can be the relatively higher contribution of the direct charge transfer to the COU transformation for ZnO than for TiO_2_. The r_0_^7-HC^ value became constant above 0.5 g dm^−3^ for both photocatalysts (6.4 × 10^−9^ mol dm^−3^ s^−1^ and 4.7 × 10^−9^ mol dm^−3^ s^−1^ for TiO_2_ and ZnO, respectively). When LED_398nm_ was applied, the r_0_^COU^ and r_0_^7-HC^ increased to 1.0 g dm^−3^ (Figure 1b). Consequently, for further experiments, 1.0 g dm^−3^ photocatalyst concentration was used.

Applying LED_365nm_, r_0_^COU^ and r_0_^7-HC^ reached the maximum at 5.0 × 10^−4^ M COU concentration and did not change significantly with the further increase (Figure 1c). In the case of LED_398nm,_ a similar trend was observed; the r_0_^7-HC^ reached the maximum value at 5.0 × 10^−4^ M COU (Figure 1d), although the r_0_^COU^ slightly increased. Consequently, for further experiments, 5.0 × 10^−4^ M COU and 1.0 g dm^−3^ catalyst concentrations were used to maximize the r_0_^7-HC^.

At constant photocatalyst (1.0 g dm^−3^) and COU (5.0 × 10^−4^ M) concentrations, the formation rate of photogenerated charges depends on the photon flux. Therefore, experiments were performed at different light intensities (Figure 1e,f). Since increasing the photon flux can be achieved by increasing the electrical energy investment, these results are also crucial in optimizing the operation cost of heterogeneous photocatalysis. Using LED_365nm_, high values of Φ_app_^COU^ and Φ_app_^7-HC^ were obtained at relatively low photon fluxes; the apparent quantum efficiency significantly decreases with the photon flux increase (Table 1), while r_0_ values change according to a saturation curve. Doubling the electrical power (6.55 → 13.60 W) and the photon flux in this way, the r_0_ values increased by only 20–30%, and the further increases (13.60 → 20.77 W) did not change that significantly for both photocatalysts. For LED_398nm_ the r_0_ values increased, and there was no significant change of the apparent quantum yield within the photon flux range applied in the case of TiO_2_. A similar trend was observed for ZnO for both LEDs (Table 1) than for TiO_2_ and LED_365nm_; the Φ_app_ value (especially the Φ_app_^7-HC^) decreased with the intensity increase.

At a given wavelength, the absorption properties and the bandgap of the photocatalyst determine the excitation efficiency primarily. In addition to these factors, the recombination rate of the photogenerated charge carriers also determines the transformation efficiency of organic substances and the radical generation rate. Based on the spectrum of the light sources and the absorbance of the catalysts (Figure 2), it can be stated that the photons of the LED_365nm_ (radiating between 355–380 nm) can be wholly absorbed by ZnO, while about 80% of photons are absorbed by TiO_2_ due to its less favorable optical properties. According to the bandgaps (3.0 eV for rutile, 3.2 eV for anatase and ZnO), both photocatalysts can be excited using this light source. Nevertheless, ZnO and TiO_2_ can absorb no more than ~20% of the emitted photons when LED_398nm_ (radiating between 385–420 nm) is used (Figure 2). The difference between absorbance of the photocatalysts at 365 and 398 nm is well reflected by the Φ_app_^COU^ values for LED_365nm_ and LED_398nm_ (Table 1).

Another critical difference is that, for TiO_2_, the photons emitted by the LED_398nm_ can primarily excite the rutile phase (Figure 2). Tang et al. [76] reported a quantum efficiency dependence on the photon flux according to a maximum curve, which correlates well with our results for LED_365nm_ (Table 1). Probably, above a given value, which value depends on electron mobility, the further increase of the photon flux enhances the recombination of the photogenerated charges, while several factors limit the redox reactions on the surface. However, Sachs et al. [77] reported that the rutile shows faster but less intensity-dependent recombination of photogenerated charges than anatase, which can be the reason for similar Φ_app_ values measured at different intensities of 398 nm light, as mainly the rutile phase is excited in this case.

The effect of irradiation wavelength is moderated when r_0_^7-HC^ and Φ_app_^7-HC^ values are compared (Table 1). It indicates that despite the lower transformation rates of COU due to the less efficient absorption of 398 nm than 365 nm photons (Figure 2), the relative contribution of ^•^OH to the transformation of COU is probably greater at 398 nm irradiation. The yield of 7-HC (r_0_^7-HC^/r_0_^COU^) is 0.02 for TiO_2_, 0.01 for ZnO when LED_365nm_ is used, and significantly higher values, 0.045 for TiO_2_ and 0.06 for ZnO, are obtained when LED_398nm_ is applied (Table 1). Presumably, due to the lower efficiency of the excitation of photocatalysts and consequently the number of photogenerated charges, the probability of e_CB^−^_–h_VB_^+^ recombination also decreases, and the quantum yield of ^•^OH formation increases. For further experiments, 6.56 W for the LED_365nm_, and 4.68 W for the LED_398nm_ were used, where the apparent quantum yields and transformation rates were relatively high.

### 3.3. The Effect of Matrices and Matrix Components on the ^•^OH Formation

The components of the treated water have a crucial role during heterogeneous photocatalysis, and their effect can be complex. The ionic components can change the surface properties of the photocatalyst (surface charge and potential), and in this way, they affect the interaction between the surface of the photocatalyst and the target compounds and the formation rate of ROS [36,37,78,79,80]. The generally negative effect of matrices is usually attributed to the scavenging of ^•^OH, the occupation of adsorption sites by the well-adsorbed inorganic and organic matrix components, or the aggregation of photocatalyst particles at higher ionic strength. Two real water matrices, tap water and biologically treated domestic wastewater (Appendix A), were used in this work to study the matrix effect on the transformation rate of COU and ^•^OH formation rate and characteristic differences between TiO_2_ and ZnO were observed. In the case of TiO_2,_ both matrices reduced the r_0_^COU^ and the r_0_^7-HC^ significantly and to a similar extent (Figure 3a,b). For ZnO, the transformation of COU was only slightly inhibited, while the r_0_^7-HC^ increased, especially in the case of LED_365nm_ (Figure 3a,b).

Due to the significant difference observed between TiO_2_ and ZnO, first, the effect of 5.0 × 10^−3^ M methanol (MeOH) as a non-absorbed ^•^OH-scavenger (k_MeOH+•OH_ = 9.7 × 10^8^ mol^−1^ dm^3^ s^−1^ [81]) was investigated. MeOH has a similar effect using both catalysts and LEDs (Appendix A). The inhibition effect on COU transformation and 7-HC formation (decrease by 45–55%) is similar and reflects well the calculated ^•^OH scavenging capacity of MeOH (about 40% of the ^•^OH reacts with MeOH at the given initial concentrations). All this suggests that the difference observed for TiO_2_ and ZnO cannot be attributed solely to the ^•^OH scavenging effect of the organic matter content of the matrices (0.79 mg dm^−3^ and 6.9 mg dm^−3^ TOC content for tap water and biologically treated domestic wastewater, respectively) (Appendix A).

The effect of the two most abundant anions of matrices, Cl^−^ and HCO_3_^−^, was investigated in suspensions containing one or both anions. Their concentrations were set to the same value as the biologically treated domestic wastewater (120 mg dm^−3^ Cl^−^ and 525 mg dm^−3^ HCO_3_^−^ (Appendix A)). In the case of LED_365nm_, Cl^−^ did not affect the r_0_^COU^ and the r_0_^7-HC^ for TiO_2_, but increased the r_0_^7-HC^ for ZnO by 63% with unchanged r_0_^COU^ (Figure 3c). Cl^−^ is reported to react with ^•^OH with a high reaction rate (k_•OH + Cl−_ = 3.0 × 10^9^ M^−1^ s^−1^ [81]); consequently, an inhibition is expected. The backward reaction reforming ^•^OH can occur at the pH values used in the current work (6.0 for TiO_2_ and 7.5 for ZnO) [36]. However, this does not explain the difference observed between the two catalysts. The adsorption of ions, the change of surface charge, and reaction with photogenerated charges also must be considered. The very slow conversion [82] and its negligible h_VB_^+^ scavenging effect of Cl^−^ in TiO_2_ suspensions were reported [38]. Opposite to the TiO_2_, the Cl^−^ are adsorbed well on the ZnO surface, having a positive charge [83,84,85] and enhancing the formation rate of ROS due to the hindered recombination of photoinduced e_CB^−^_ and h_VB_^+^ [83]. Significant conversion of Cl^−^ by h_VB_^+^ to HClO has been reported in some cases [82,86,87]. The degradation of HClO via reaction with e_CB^−^_, or superoxide radical anions (O_2^•−^_) also results in the formation of ^•^OH [87,88], which may explain the increased r_0_^7-HC^ for ZnO. Active chlorine species formation and photocatalytic reactions are complex, depending on many parameters [88,89,90], and require further investigation. Moreover, the positive effect of Cl^−^ was observed only in the case of 365 nm radiation. This suggests that the excess energy of the photons or the concentration of photogenerated charges can have a role in the manifestation of the Cl^−^ effect in the case of ZnO.

The HCO_3_^−^ reduced the r_0_^COU^ and r_0_^7-HC^ by the same extent (by about 30%) when ZnO and LED_365nm_ were used. For TiO_2_, the r_0_^COU^ was reduced by 50%, but r_0_^7-HC^ was more than doubled (Figure 3c). Due to the relatively low reactivity of HCO_3_^−^ with ^•^OH (k_•OH + HCO3−_ = 1.0 × 10^7^ M^−1^ s^−1^ [81]), the inhibition of COU transformation cannot be attributed solely to its ^•^OH scavenging role (~5% of ^•^OH reacts with HCO_3_^−^); its reaction with h_VB_^+^ is also essential. The reaction between HCO_3_^−^ and h_VB_^+^ results in CO_3^•−^_ formation only in the case of TiO_2_, not in the case of ZnO [38,40]. The formed CO_3^•−^_ is a selective and less reactive reaction partner than ^•^OH, but its reaction with aromatic compounds results mainly in hydroxylated products [91,92,93], similar to ^•^OH. Thus, the formation of CO_3^•−^_ instead of ^•^OH can be responsible for the r_0_^COU^ decrease and probably for the r_0_^7-HC^ increase. These results also indicate that r_0_^7-HC^ is unsuitable for determining the ^•^OH formation rate when CO_3^•−^_ forms in the system. Comparing the effect of HCO_3_^−^ in the 365 and 398 nm irradiated TiO_2_ suspensions, the inhibition of the COU transformation is more pronounced in the case of 398 nm radiation, likely because of the less efficient excitation; consequently, there is a lower concentration of both photogenerated charges and ^•^OH. The difference between the two light sources is exciting, as additives have a characteristic effect on the formation of 7-HC in the case of LED_365nm_.

The change of the mineralization rate confirmed the enhanced formation rate of less reactive CO_3_^•−^ instead of ^•^OH. In the case of 365 nm radiation, the mineralization rate was the same for TiO_2_ and ZnO, and significantly reduced by HCO_3_^−^ to a much greater extent for TiO_2_ than for ZnO (Figure 4a). The reaction between HCO_3_^−^ and h_VB_^+^, and the formation of CO_3_^•−^, is likely to play a significant role only in the case of TiO_2_. In the case of ZnO, the HCO_3_^−^ primarily acts as an ^•^OH scavenger, while in the case of TiO_2_ it also reacts with ^•^OH and the h_VB_^+^. The formed CO_3_^•−^ is less efficient for mineralization than ^•^OH. Thus, mineralization is inhibited by HCO_3_^−^ to a greater extent for TiO_2_ than for ZnO. The Cl^−^ enhanced the r_0_^7-HC^ for ZnO but had no significant effect on the r_0_^COU^ (Figure 3c) and mineralization (Figure 4b). One possible interpretation is that the radical scavenging effect of Cl^−^ and its positive effect on charge separation and/or regeneration of ^•^OH from Cl-containing species compensate for each other.

A significant difference was observed between mineralization rates for ZnO, and TiO_2_ determined in the suspensions irradiated with 398 nm light, although there was no difference in the case of 365 nm irradiation. However, 398 nm light excites only the rutile phase, and the TiO_2_ is much more effective in mineralization than ZnO having slightly better optical properties at a longer wavelength range. Moreover, the inhibition effect of HCO_3_^−^ is manifested only for TiO_2_ (Figure 4c).

In suspensions containing both Cl^−^ and HCO_3_^−^, the synergistic effect of the anions was not observed. In the case of 365 nm radiation, the negative effect of HCO_3_^−^ on r_0_^COU^ was dominant. At the same time, for ZnO, the Cl^−^ effect while for TiO_2_ the HCO_3_^−^ effect was dominant on the r_0_^7-HC^ (Figure 3c). The extent of the effect depended on the wavelength; in the case of LED_398nm_, only a slight reduction of reaction rates was observed (Figure 3d). Comparing the effect of anions and their mixtures to the tap water and biologically treated domestic wastewater (Figure 3a,b), the reduction of photocatalytic activity in these matrices cannot be solely responsible for the presence of inorganic anions, especially for TiO_2_. For ZnO, the inhibition effect matrices are similar to that of the Cl^−^ and HCO_3_^−^ mixture; the r_0_^COU^ slightly decreased, in the case of both LEDs, while r_0_^7-HC^ increased, especially for 365 nm radiation. However, for TiO_2,_ the transformation is much more inhibited and cannot be explained by the effect of Cl^−^ and HCO_3_^−^ and the relatively low organic content of the matrices.

### 3.4. Removal Efficiency of Neonicotinoids

The photocatalytic removal of two neonicotinoid pesticides, IMIDA and THIA, with severe environmental impact was also investigated. As expected, the LED_365nm_ was much more effective; nearly complete removal (>90%) was achieved within 30 min for both pesticides while using LED_398nm_ this time increased to 90–120 min (Figure 5a,b). A slightly higher transformation rate was observed for ZnO than TiO_2_ using LED_398nm_ (Figure 5a,b). The apparent quantum efficiency of LED_398nm_ (5.3–7.9 × 10^−4^) was only 10–20% of the values determined for LED_365nm_ (4.8–8.5 × 10^−3^), with a good agreement of the number of absorbed photons, estimated from the absorption of the catalysts and the emission spectra of the LEDs (Figure 1).

The main goal of AOPs is generally the complete removal of harmful organic compounds and their degradation products. While in the case of COU, there was no difference between the efficiency of the two catalysts using LED_365nm_, faster mineralization of both neonicotinoids was observed for TiO_2_ than for ZnO (Figure 5c,d). Using LED_365nm_, 10–20% of TOC could not be removed even after 120 min, which indicates the formation of degradation products resistant to photocatalytic treatment. These products most likely formed in more significant amounts in the case of ZnO, when more than 40% of TOC remained in the suspension after 60 min and decreased very slowly during further treatment. The observed difference can be interpreted by the higher ^•^OH formation rate for TiO_2_ and probably by the higher contribution of the direct charge transfer to the transformation of neonicotinoids and their intermediates for ZnO. In addition, opposite to the neonicotinoids, TOC decreased linearly during the whole treatment of COU (Figure 4 and Figure 5c,d).

Since one of the main arguments in favor of LEDs is their better consumption of electric power, the energy required to treat a unit volume of IMIDA and THIA suspension was compared. In the case of LED_398nm_ (4.68 W), the energy needed to reduce the concentration by 90% [94] was 5–6 times higher for THIA, and 7–9 times higher for IMIDA than in the case of LED_365nm_ (6.56 W) (Figure 5e). Comparing the electrical energy required to eliminate the 50% of TOC content, the difference between the two LEDs was moderate; for ZnO about four times higher, and TiO_2_ about five times higher energy was needed (Figure 5f). Despite the higher power usage and slightly lower electrical efficiency (Table 1 and Appendix A), the higher energy of the emitted photons made the LED_365nm_ economically more favorable due to the better optical properties of both photocatalysts at 365 nm than at 398 nm.

For IMIDA, four degradation products were identified (Figure 6); IM/1 (m/z = 230.0) resulted by the opening of imidazolidine ring [95], while the attack of the ^•^OH to the imidazolidine and –CH_2_– moiety led to the formation of IM/2 (m/z = 272.1), IM/4 (m/z = 270.1), and IM/3 (m/z = 288.0). The products distribution differed for TiO_2_ and ZnO; IM/1 and IM/2 formed significantly faster using ZnO (Appendix A), confirming a different relative contribution of ^•^OH based reactions and direct charge transfer to IMIDA transformation for TiO_2_ and ZnO. For THIA, the hydroxylation of the thiazolidine ring and –CH_2_– moiety led to the formation of T3 (m/z = 269.1) and T/4 (m/z = 269.1) (Figure 6). Their further transformation resulted in carbonylated T/2 (m/z = 267.0) and hydroxylated T/5 (m/z = 285.1) products. In the case of THIA, the product distribution was similar for both photocatalysts (Appendix A). 

The mineralization results in inorganic ions depend on the chemical structure of the pollutants [96]. The dechlorination was complete with both photocatalysts after 60 min using LED_365nm_ and reached 90% using LED_398nm_ after 120 min (Figure 7a). The S-content was converted to SO_4_^2−^, and its accumulation was much slower than that of Cl^−^, as it required the ring-opening process of the thiazolidine ring. Less than 50% of S-content was converted to SO_4_^2−^ after 60 min and reached 74% at 120 min treatment using LED_365nm_. This value only reached 20–30% using LED_398nm_ (Figure 7b), following the slower TOC removal.

The fate of organic N-content depends on the oxidation state of the N atom in the molecule, on its chemical environment, and the reaction parameters [96,97]. N-containing moiety can transform into NO_3_^−^, NH_4_^+^ and N_2_ as final products. Toxic NO_2_^−^ may also form but is usually oxidized to NO_3_^−^ by ^•^OH (k_•OH + NO2−_ = 6.0 × 10^−9^ M^−1^ s^−1^ [98]), while oxidizing NH_4_^+^ to NO_3_^−^ is very slow. The conversion of N-content of THIA to NH_4_^+^ (24%) exceeded that determined for IMIDA (14%), and TiO_2_ resulted in significantly higher concentrations of NH_4_^+^ compared to ZnO (Figure 7c). IMIDA produces significantly higher NO_3_^−^ concentrations than THIA, as the –NO_2_ group is likely to be easily converted to NO_3_^−^ (Figure 7d). The NO_2_^−^ concentration is negligible for TiO_2_, but 5–8% and 3–4% of the total N-content can be detected as NO_2_^−^ during the transformation of IMIDA and THIA, respectively, when ZnO was used (Figure 7e). Park et al. reported the reduction of NO_3_^−^ on ZnO, which does not occur on TiO_2_ [99]. This can explain the presence of NO_2_^−^ but also points out that it can be a disadvantage in removing N-containing organic impurities with ZnO. The NO_2_^−^ is hazardous to the environment and human health, even in low concentrations (1–3 mg dm^−3^).

### 3.5. Effect of Matrices on the Removal of IMIDA and THIA

For TiO_2,_ the transformation rates of both IMIDA and THIA were significantly reduced in real matrices using both light sources, while for ZnO, the matrices did not affect the removal rates or even increase them (Figure 8a,b). Cl^−^ had no effect in the case of TiO_2_, but enhanced the transformation rate significantly when ZnO was used (Figure 8c,d). In the case of COU transformation, the positive effect on r_0_^7-HC^ was explained by the enhanced charge separation or the formation of reactive Cl-containing species, but the unchanged COU mineralization rate did not confirm the enhanced ^•^OH production. In the case of neonicotinoids, the positive effect can be explained by the direct reaction of neonicotinoids with formed reactive chlorine species, as Yin et al. [100] reported a significant reactivity of IMIDA and THIA towards various reactive chlorine species. However, the effect of HCO_3_^−^ is remarkably different for COU and neonicotinoids: HCO_3_^−^ inhibited the COU transformation (Figure 3c,d), but it had no significant effect on the transformation rates of IMIDA and THIA (Figure 8c,d), even in the case of TiO_2_.

To gain a deeper understanding, we examined the effect of inorganic ions on the formation of intermediates, and significant differences were observed, despite the minor effect on the removal rates. In the case of IMIDA, the Cl^−^ significantly increased the formation rate of the hydroxylated product (IM/3) using both photocatalysts (Appendix A), while HCO_3_^−^ increased the formation of IM/1 and IM/2 products. Dell’Arciprete et al. studied the reactions of IMIDA with CO_3_^•−^ and presented hydroxylated products and products formed via the opening of the imidazolidine ring [92]. For THIA, Cl^−^ had no effect, but HCO_3_^−^ inhibited the formation of T/2 and T/4 (Appendix A). It should be mentioned that CO_3_^•−^ reacts significantly more slowly with neonicotinoids (k_IMIDA + CO3•−_ = 4.0 × 10^6^ M^−1^ s^−1^; _kTHIA+ CO3•−_ = 2.8 × 10^5^ M^−1^ s^−1^ [92]) than ^•^OH (k_IMIDA+•OH_ = 7.0 × 10^9^ M^−1^ s^−1^; k_THIA+•OH_ = 4.8 × 10^9^ M^−1^ s^−1^ [101,102]), but due to the longer lifetime and selectivity, CO_3_^•−^ may be present at higher concentration, and in the case of TiO_2_ it can contribute significantly to the transformation [103].

The potential of the valence band of TiO_2_ and ZnO creates the possibility of highly reactive radical formation due to the reaction with h_VB_^+^. The effect of inorganic ions is complex during heterogeneous photocatalysis and often depends on pH via acid–base equilibrium processes. Ions adsorbed on the surface can change the surface charge and potential of valance and conduction band edge [104]. Moreover, they can react directly with the photogenerated charges—which means a competition for their reaction with H_2_O/HO^−^ resulting in ^•^OH. The selectivity and reactivity of the formed radicals or radical ions are different. The electrode potential of the inorganic radical ions is a powerful indicator of their reactivity; thus, it is worth comparing their standard potentials [105] (Appendix A). Comparing these values, we can state that the CO_3_^•−^ forms easily, while competition can occur between the formation of ^•^OH and Cl^•^, which is influenced by pH. Because of the complexity of the processes, the correct interpretation of the effect of various inorganic ions on the radical set requires further investigations, even in the case of TiO_2_ and ZnO suspensions.

### 3.6. Ecotoxicity Change

The toxicity change was investigated during photocatalytic treatment using *Vibrio fischeri* as a test organism (Figure 9). The toxicity of THIA solution (44% inhibition) was significantly higher than IMIDA (20–25% inhibition). Using LED_365nm_, the toxicity of THIA quickly reduced; after 90% TOC removal, no inhibition of bioluminescence was detected. For IMIDA, the toxicity slightly increased during the first 30 min, then slowly decreased, but remained significant even after 80% reduction of TOC (Figure 5a,c). Using LED_398nm_, the toxicity changed more slowly, as expected based on the less effective transformation and mineralization rates (Figure 5b,d); however, the trends were similar to those observed for LED_365nm_. During the transformation of IMIDA, toxic products form [106], despite the 90% removal of TOC. The significant difference between the conversions of these neonicotinoids is the formation of NO_3_^−^, which is characteristic only of IMIDA. The formation of nitro products during the treatment of N-containing organic contaminants is generally a consequence of the reaction with reactive nitrogen species formed from NO_2_^−^ or NO_3_^−^. The nitro-derivatives are often more toxic compounds than the primary pollutants, which could be the reason for increased toxicity in IMIDA transformation.

Although ZnO is less effective in removing TOC than TiO_2_ at 365 nm irradiation (Figure 5), there is no significant difference in the time dependence of the toxicity change, especially for THIA (Figure 9). It is worth mentioning that while in the case of LED_365nm_ the toxicity of the THIA solution disappears after 90 min, in the case of LED_398nm_ the toxicity just slightly decreases and practically does not change after 60 min treatment, opposite that during 120 min treatment almost 50% of TOC eliminated (Figure 5). 

## 4. Conclusions

During the last decades, the intensive development of LED technology, especially high-intensity LEDs, created an opportunity to replace traditional UV light sources in water treatment processes. In this work, we compared the efficiency of a high-power LED_365nm_ and a commercial, low-cost LED_398nm_ in the case of heterogeneous photocatalysis using TiO_2_ and ZnO photocatalysts. The comparison was based on the rate of transformation and mineralization of various organic substances, the ^•^OH formation rate, the energy consumption, and the matrix effect, with particular attention to the effect of its inorganic components such as Cl^−^ and HCO_3_^−^.

The application of high-power LED_365nm_ was more economical at the lower electrical power and light intensity; the apparent quantum yield decreased with increasing light intensity, while the conversion rate varied according to the saturation curve. Mainly because of the optical properties of the photocatalysts, the transformation of COU was significantly slower for LED_398nm_ than for LED_365nm_, but the yield of 7-HC (r_0_^7HC^/r_0_^COU^) was significantly higher when LED_398nm_ was applied. The mineralization rate of COU was the same for both catalysts at 365 nm irradiation but differed significantly at 398 nm irradiation.

The impact of matrices Cl^−^ and HCO_3_^−^ was completely different for ZnO and TiO_2_. For ZnO, the Cl^−^ significantly increased the formation rate of r_0_^7-HC^, but no change of r_0_^COU^ was observed. The negative effect of HCO_3_^−^ as radical scavenger and hole trapping species was evident for both catalysts; however, in the case of TiO_2_, the formation of CO_3_^•−^ is almost double the 7-HC formation rate. The inhibitory effect of real matrices was much more significant for TiO_2_ and cannot be interpreted by the combined effect of these ions even in tap water with low organic matter content. However, for ZnO, the effect of matrices was negligible or even positive.

Two relevant environmental pollutants, IMIDA and THIA, showed similar transformation rates for TiO_2_ and ZnO, but TiO_2_ was more favorable in mineralization. In both cases, hardly oxidizable products were formed, but to a significantly greater extent for ZnO, most likely because of the higher contribution of the direct charge transfer to the transformation. The formed CO_3_^•−^ likely contributes to the conversion of these compounds, especially in the case of TiO_2_. In the case of IMIDA, the increased toxicity is presumably due to reactions with N-containing reactive species formed from the NO_2_^−^/NO_3_^−^.

The better absorption properties of TiO_2_ and ZnO photocatalysts at 365 nm made the LED_365nm_ economically more favorable than LED_398nm_. However, the application of these catalysts modified to absorb a higher percentage of the 398 nm photons may greatly increase the efficiency of LED_398nm_. The significantly lower price, slightly higher electrical efficiency of LED_398nm_, and relatively good ^•^OH generation ability of 398 nm light, may provide an alternative for using LED_398nm_ to eliminate the hazardous organic matter by heterogeneous photocatalysis. Our results showed that the two most commonly used catalysts, TiO_2_ and ZnO, react differently with the inorganic ions, affecting the efficiency in a complex way in a real matrix. The secondary reactive species formation from inorganic ions and their role and effect depend on the target compound and the photocatalyst. In addition to changing the conversion rate of the starting compound, the analysis of the products formed also provides essential information about the processes taking place.

## Figures and Tables

**Figure 1 nanomaterials-12-00005-f001:**
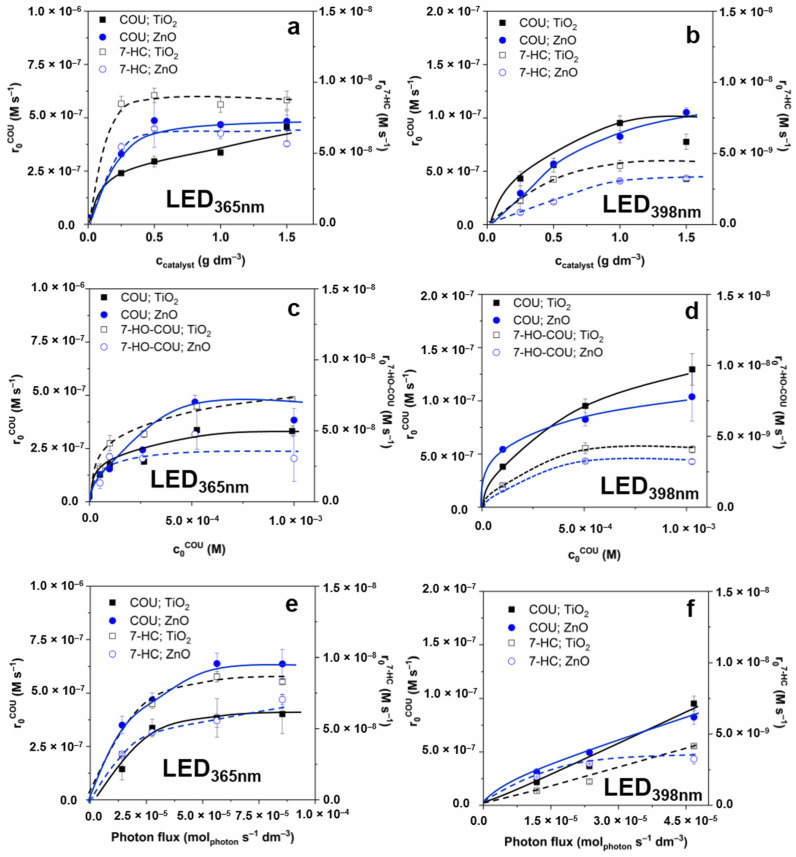
The effect of catalyst load (**a**,**b**), COU initial concentration (**c**,**d**), and photon flux (**e**,**f**) on the initial transformation rate of COU and 7-HC.

**Figure 2 nanomaterials-12-00005-f002:**
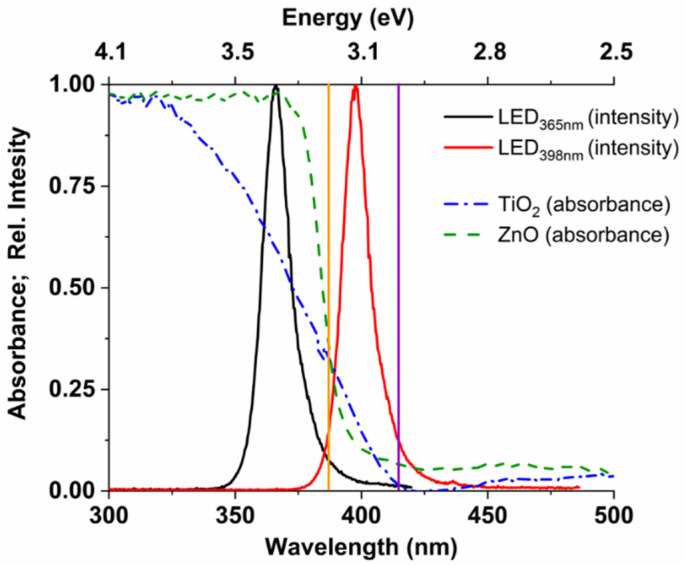
The emission spectra of the LEDs (black and red line), and the absorption spectra of the TiO_2_ and ZnO photocatalysts (blue and green interrupted line). The bandgap of the ZnO and TiO_2_ anatase (3.2 eV) with a vertical orange line, and the bandgap of TiO_2_ rutile (3.0 eV) with a vertical purple line).

**Figure 3 nanomaterials-12-00005-f003:**
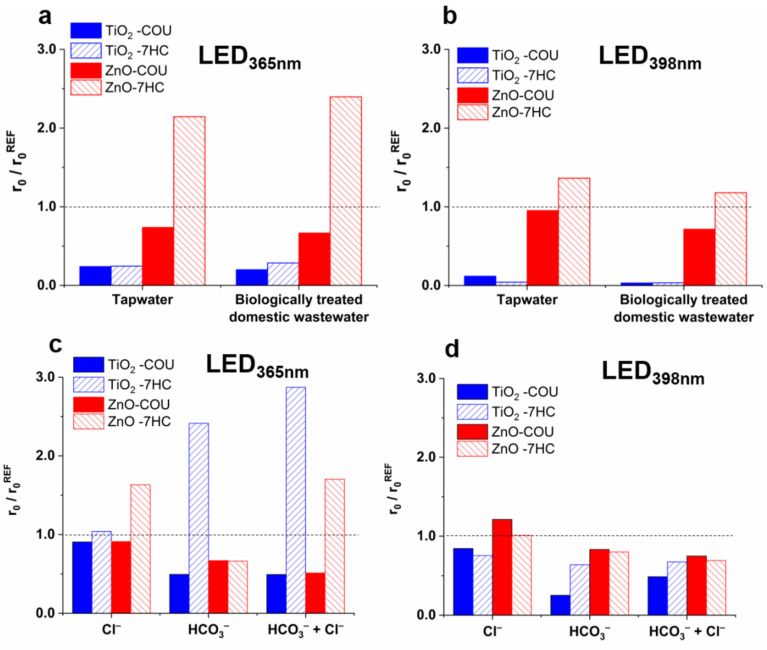
Effect of matrices (**a**,**b**), Cl^−^ (120 mg dm^−3^), and HCO_3_^−^ (525 mg dm^−3^) (**c**,**d**) on the relative transformation rate of COU and formation rate of 7-HC.

**Figure 4 nanomaterials-12-00005-f004:**
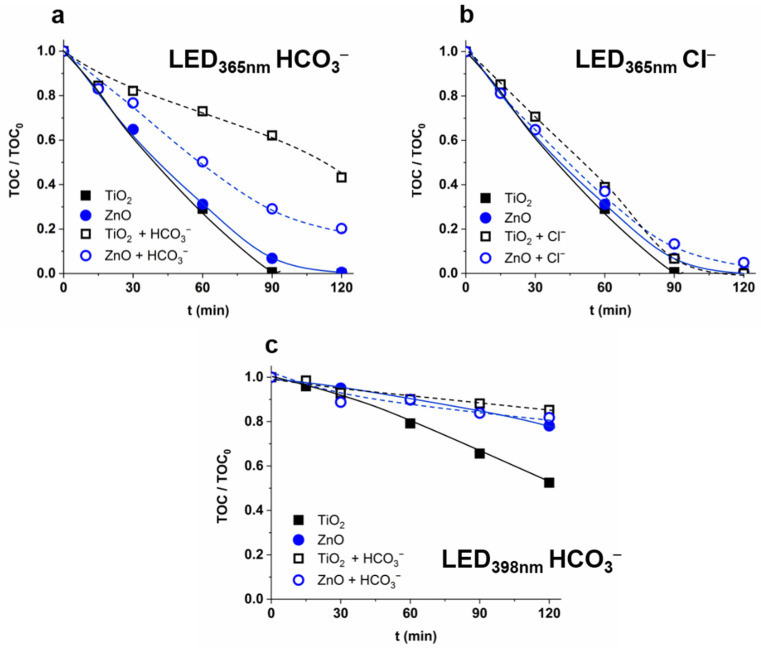
The effect of HCO_3_^−^ (525 mg dm^−3^) (**a**,**c**) and Cl^−^ (120 mg dm^−3^) (**b**) on the mineralization of COU (c_0_ = 5.0 × 10^−4^ M; TOC_0_ = 55 mg dm^−3^).

**Figure 5 nanomaterials-12-00005-f005:**
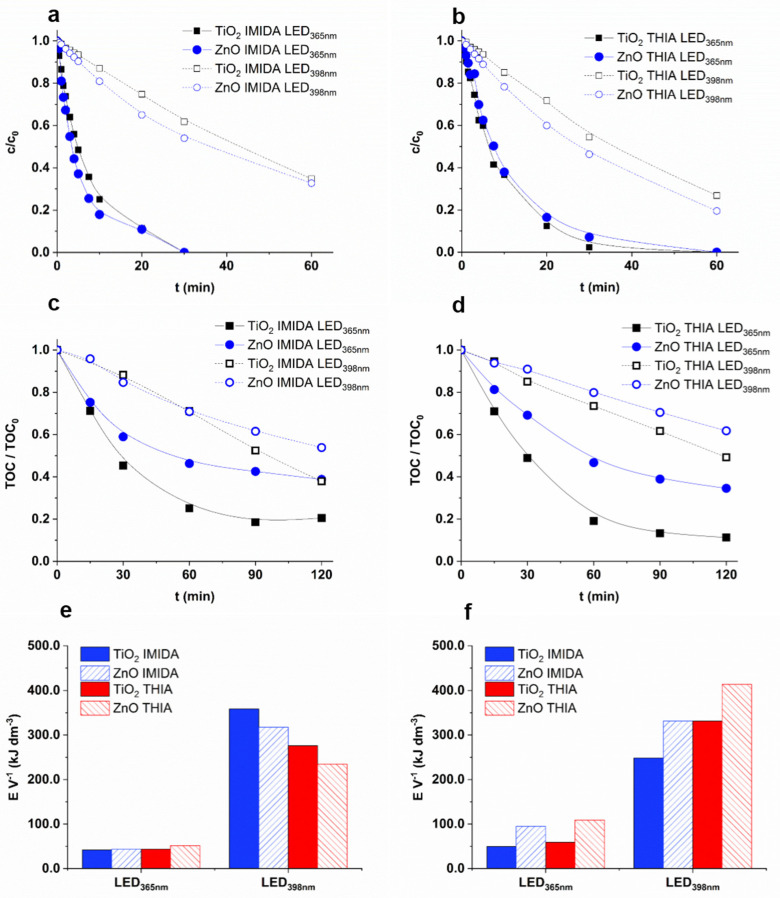
The relative concentration (**a**: IMIDA; **b**: THIA), and the relative TOC content (TOC_0_ = 12 mg dm^−3^) (**c**: IMIDA; **d**: THIA) as a function of irradiation time, and the electric energy required to remove 90% of the neonicotinoid (**e**) and 50% of initial TOC content (**f**).

**Figure 6 nanomaterials-12-00005-f006:**
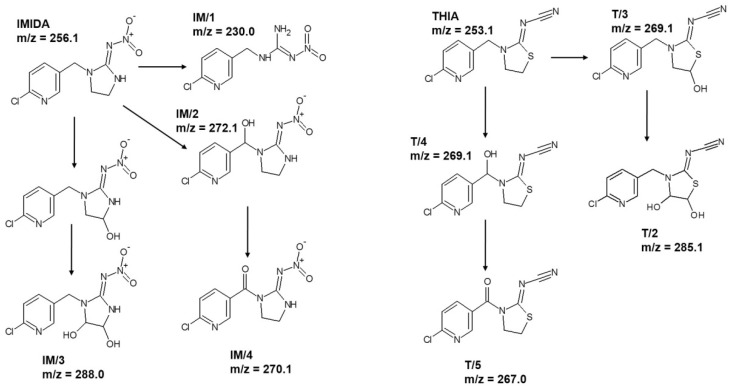
The products of IMIDA and THIA detected by HPLC-MS.

**Figure 7 nanomaterials-12-00005-f007:**
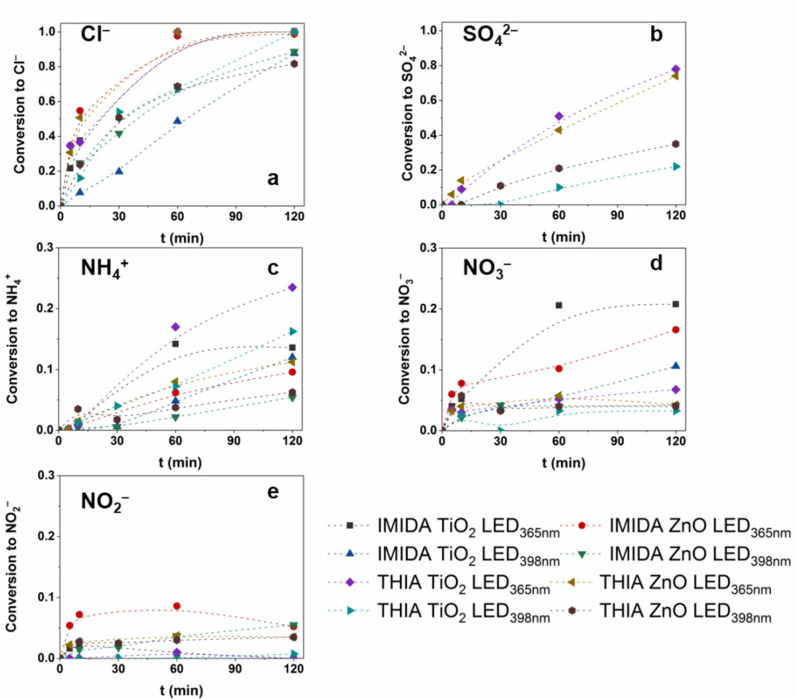
The conversion of organic Cl-, S-, and N-content of IMIDA and THIA to Cl^−^ (**a**), SO_4_^2−^ (**b**), NH_4_^+^ (**c**), NO_3_^−^ (**d**) and NO_2_^−^(**e**).

**Figure 8 nanomaterials-12-00005-f008:**
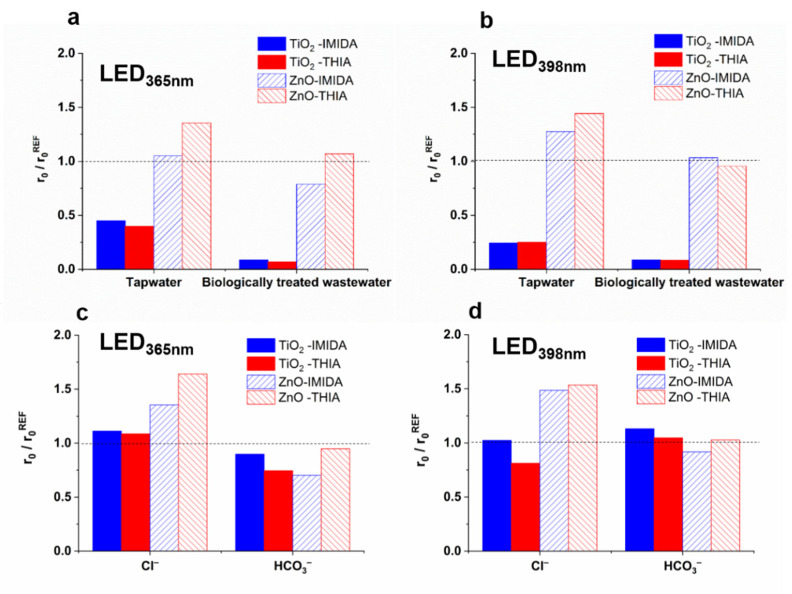
The relative initial transformation rates of IMIDA and THIA measured in different water matrices (**a**,**b**), and in the presence of Cl^−^ and HCO_3_^−^ (**c**,**d**).

**Figure 9 nanomaterials-12-00005-f009:**
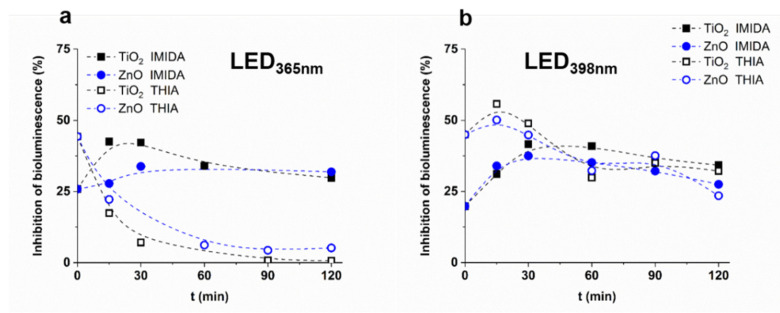
The ecotoxicity of IMIDA and THIA suspensions as a function of treatment time ((**a**): LED_365nm_, (**b**): LED_398nm_).

**Table 1 nanomaterials-12-00005-t001:** The apparent quantum yields (Φ_app_) of the transformation of COU and the formation of 7-HC (c_0_^COU^ = 5.0 × 10^−4^ M and 1.0 g dm^−3^ ZnO or TiO_2_).

LED	P_el_ (W)	TiO_2_					ZnO				
		r_0_^COU^ (×10^−7^)	Φ_app_^COU^ (×10^−2^)	r_0_^7-HC^ (×10^−9^)	Φ_app_^7-HC^ (×10^−4^)	r_0_^7-HC^/r_0_^COU^	r_0_^COU^ (×10^−7^)	Φ_app_^COU^ (×10^−2^)	r_0_^7-HC^ (×10^−9^)	Φ_app_^7-HC^ (×10^−4^)	r_0_^7-HC^/r_0_^COU^
**LED_365nm_**	3.39	1.49	1.01	3.4	2.3	0.023	3.50	2.47	3.22	2.3	0.009
	6.56	3.03	1.22	6.40	2.4	0.021	4.68	1.70	4.75	1.7	0.010
	13.60	3.83	0.68	8.25	1.5	0.022	6.21	1.13	5.57	1.0	0.009
	20.77	4.02	0.47	8.30	1.0	0.021	6.20	0.75	6.25	0.8	0.011
**LED_398nm_**	0.96	0.22	0.18	1.02	0.85	0.044	0.31	0.26	2.00	1.7	0.039
	2.16	0.47	0.19	1.97	0.82	0.045	0.50	0.21	2.93	1.3	0.059
	4.68	0.95	0.20	4.15	0.89	0.047	0.83	0.18	3.25	0.7	0.064

Φ_app_: the transformation (**r_0_^COU^**) or formation rate (**r_0_^7HC^**) divided by the incoming photon flux.

## Data Availability

The data is included in the article or Appendix A.

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
