# Peer review of "Impact of Reaction Parameters and Water Matrices on the Removal of Organic Pollutants by TiO2/LED and ZnO/LED Heterogeneous Photocatalysis Using 365 and 398 nm Radiation"

_nanomaterials, 2021, doi:10.3390/nano12010005_

Round 1
Reviewer 1 Report
Please see the attached file

Author Response
Respected reviewer,
Thank you for accepting the review request and for improving the quality of the manuscript with your comments. Thanks again for the positive feedback and suggestions.
Figure 2 and Figure 6 are corrected (see the attached file). We also developed the grammar and spelling of the manuscript.
"Please see the attachment."

Reviewer 2 Report
The manuscript describes the application of low-price LED for photocatalytic degradation of water pollutants using TiO2 and ZnO. The manuscript is well structured and well written. I recommend the manuscript for publication in this journal.
Author Response
Respected reviewer,
Thank you for accepting the review request. Many thanks for the positive feedback and recommendation of the manuscript for publication.
Reviewer 3 Report
The main topic of this paper is a comparative evaluation of different photocatalysts such as TiO2 and ZnO for OH radicals formation using different UV-LED light source such as 365 and 398 nm. The manuscript in general is well done and easy to follow. However, the research topic is not novel and there were many scientific errors in the experimental method. I wonder why you used COU to check OH radicals rate instead of using usual probe compound such as pCBA. Also, there seemed to be a lack of justification for that. You used a light source in the UV-A wavelength, but did not apply quartz to the reactor, but designed it with glass. I didn't quite understand this part. In addition, the UV intensity of 365 nm and 398 nm were measured with the same detector and analyzed in the same actinometry method, but no correction values ​​for each wavelength band or light blocking filter were used. Also, the fact that the reactors with the each UV-LED lamps from two different sources are different can also be a problem. Unfortunately, I regret to inform you that my decision on this manuscript is a reject.
Author Response
Respected Reviewer,
Thank you for accepting the review request, your work, and your comments.
The answers can be found in the attachment file. We sincerely hope that based on our detailed responses, the reviewer will consider the opinion and decision.
"Please see the attachment."

Reviewer 4 Report
The paper compares the efficiency of two photocatalysts, ZnO and TiO2, in removal of organic pollutants under different conditions of illumination. The aspect in which authors analyze their results is indeed very interesting and of great importance from the practical point of view. The paper is well written and presents a thorough analysis of the systems, which are very complex and straightforward conclusions may not always be drawn. I recommend publication of this paper after minor revision. My suggestions, which might be of relevance, are the following.
- XRD analysis of TiO2 would answer the question regarding phase composition of the photocatalyst (anatase vs rutile).
- The authors could analyze and reconsider their results from the viewpoint of energetics of possible processes. The deep valence band position of TiO2 and ZnO (> 3 V (SHE) Mater., 2012, 24, 3659–3666) renders high oxidizing power to photoinduced holes. This is why they can drive oxidation reactions via pathways involving formation of highly reactive species such as radicals. The standard potentials of various radical formation can be found at Pure Appl. Chem. 87 (2015) 1139–1150. Comparison of E0 values for OH*, Cl* and CO3* radical formation clearly shows that formation of CO3* is the most facile process, whereas competition between formation of OH* and Cl* is strongly influenced by pH. I strongly believe that such analysis could help understanding the photocatalytic phenomena occurring in the systems investigated.
Author Response
Respected Reviewer,
Thank you for accepting the review request and for improving the quality of the manuscript with your comments. The suggested publications were studied and the section on radical formation was developed based on them. Thanks again for the positive feedback and suggestions. The detailed answers can be found in the attached file.
"Please see the attachment."

Round 2
Reviewer 3 Report
The authors did a good job of defending and answering against my comments.